# The Determination of On-Water Rowing Stroke Kinematics Using an Undecimated Wavelet Transform of a Rowing Hull-Mounted Accelerometer Signal

**DOI:** 10.3390/s24186085

**Published:** 2024-09-20

**Authors:** Daniel Geneau, Drew Commandeur, Ryan Brodie, Ming-Chang Tsai, Matt Jensen, Marc Klimstra

**Affiliations:** 1School of Exercise Science, Physical and Health Education, University of Victoria, Victoria, BC V8P 5C2, Canada; dcommand@uvic.ca (D.C.); mklimstra@csipacific.ca (M.K.); 2Canadian Sport Institute Pacific, Victoria, BC V9E 2C5, Canada; rbrodie@csipacific.ca (R.B.); mtsai@csipacific.ca (M.-C.T.); mjensen@csipacific.ca (M.J.)

**Keywords:** rowing, IMU, sport, machine learning, waveform

## Abstract

Boat acceleration profiles can provide valuable information for coaches and practitioners to make meaningful technical interventions and monitor the determinants of success in rowing. Previous studies have used simple feature detection methods to identify key phases within individual strokes, such as drive onset, drive time, drive offset and stroke time. However, based on skill level, technique or boat class, the hull acceleration profile can differ, making robust feature detection more challenging. The current study’s purpose is to employ the undecimated wavelet transform (UWT) technique to detect individual features in the stroke acceleration profile from a single rowing hull-mounted accelerometer. In this investigation, the temporal and kinematic values obtained using the AdMos^TM^ sensor in conjunction with the UWT processing approach were strongly correlated with the comparative measures of the Peach™ instrumented oarlock system. The measures for stroke time displayed very strong agreeability between the systems for all boat classes, with ICC values of 0.993, 0.963 and 0.954 for the W8+, W4− and W1x boats, respectively. Similarly, the drive time was also very consistent, with strong to very strong agreeability, producing ICC values of 0.937, 0.901 and 0.881 for the W8+, W4− and W1x boat classes. Further, a Bland–Altman analysis displayed little to no bias between the AdMos^TM^-derived and Peach™ measures, indicating that there were no systematic discrepancies between signals. This single-sensor solution could form the basis for a simple, cost-effective and accessible alternative to multi-sensor instrumented systems for the determination of sub-stroke kinematic phases.

## 1. Introduction

The standard for measurement in the sport of rowing is to use multi-sensor systems, such as the Peach Innovations Powerline^TM^ (Peach^TM^), in which the boat is instrumented with force, displacement and speed measurement devices on the hull, oarlocks and footplates (stretchers) [1,2,3,4,5]. With data from this comprehensive multi-sensor setup, the rowing stroke technique can be segmented into important phases based on different aspects of each measured signal. For example, it is common to use the handle angle or velocity to segment the drive and recovery based on the maximum and minimum handle excursion or velocity zero crossing [2,6]. Additionally, kinematic features of the rower, hull or handle, measured by different sensors, can further define subphases, such as initial boat or rower acceleration and catch preparation [6]. While a multi-sensor system is ideal and is required for a detailed technical analysis, cost and technical limitations preclude its broadscale use. Further, the need for a subphase analysis in rowing is not required on a daily basis, and less detailed phase information, such as simply determining stroke timing or drive and recovery phases, could provide sufficient information for coaches and athletes to support daily training decisions. To support cost-effective measurement needs in rowing, Inertial Measurement Units (IMUs) have been shown to be capable of determining rowing kinematics using a single hull-mounted IMU [2,6]. This level of phase determination would make IMUs a cost-effective and accessible tool for rowing measurement. However, a single hull-mounted IMU does not provide the same direct measurements as the multi-sensor system to determine important phase transitions, and this potentially makes the identification of phases and technical features more challenging.

The determination of stroke kinematics using hull acceleration can be aided by the existence of stereotypical patterns of acceleration during the rowing stroke that are clearly distinguished across boat classes and rowing styles [7]. A repeating peak of negative acceleration occurs during the transition from recovery to drive and can be easily collected using a hull-mounted accelerometer and detected using any peak detection algorithm [7]. While this signal feature is not temporally aligned with the start of the drive based on oar kinematics, it is a consistent component of the first subphase of the drive and is representative of definitive change in boat acceleration. Therefore, this negative acceleration has been used to determine the drive start [2,6] and also aligns with the rising threshold of oarlock force used by the Peach™ system for the calculation of power. While this feature of the acceleration signal is clearly distinguishable and easily identified by using basic signal processing, there are other features related to important transitions, such as the end of the drive and recovery, that are not easily and clearly determined using simple signal processing methods [2,6]. For example, Holt et al. (2021) identified three peaks and three valleys in a rowing hull acceleration signal that represent important variables or phase transitions [2]. They used a logical iterative algorithm, during post-processing of a filtered signal, to identify the appropriate phase transitions. For example, the end of the drive, as defined by Holt et al. (2021), was marked as the first valley after the location of the peak drive acceleration where acceleration increased and was less than 2.45 $m/s^{2}$ [2]. While Holt et al. (2021) examined many boat classes in their study, they needed to modify their signal processing techniques to appropriately determine the complementary features in the signal for their analysis and set thresholds for filtering and signal magnitude to consistently segment the data [2]. While this approach is suitable for research purposes, there is great potential for incorrectly identifying or missing important features in the signal, especially when including different boat classes or levels of rowing experience. Further, the thresholds chosen could strongly depend on the level of filtering of the signal. Based on this current standard of analysis, there is a great need to investigate signal processing techniques to improve the reliability of signal feature detection in rowing and enhance the potential for the use of hull-mounted IMUs for effective rowing analysis. 

A unique signal processing technique that has been successfully employed for time-series sports data is the undecimated wavelet transform (UWT), also known as the stationary wavelet transform [8,9,10]. This technique is a variation of the discrete wavelet transform that does not involve downsampling and uses zero-phase filters, thus allowing the transformed signal to remain time-aligned with the original signal and maintain resolution [10]. Ultimately, the UWT decomposes a signal into different frequency bands with an increasing number of levels of decomposition, creating narrower frequency bands in the resulting coefficients which can help isolate the specific features of the signal [10]. This means that the UWT can provide a detailed, frequency-dependent analysis of a signal without losing temporal information. For example, Zheng et al. (2022) applied the UWT to sports dance motion capture data before applying machine learning for movement prediction [8]. They chose to use the UWT to ensure that the signal’s movement characteristics were preserved for each signal coefficient and could identify different features in different signal-level coefficients [8]. As the rowing hull acceleration collected from an IMU contains movement features that may be better identified in different frequency bands, the UWT may be a valuable signal processing technique. Specifically, an important signal feature that could benefit from the UWT is the consistent determination of the finish (end of drive). While Holt et al. (2021) developed an algorithm to support the determination of the drive end, it is possible that the UWT may provide another suitable approach to identify this feature that requires fewer logical iterative processes and offers more reliable determination across boat classes [2]. Further, hull acceleration and force-segmented kinematic measures have never been compared, presenting a noticeable gap within the literature [2,6,11].

Therefore, this study’s purpose is to develop an approach using the UWT to accurately determine the drive start and drive end across rowing classes using acceleration data from a single rowing hull-mounted IMU signal. The drive variables determined by using the novel UWT approach will be compared to the drive start and end generated from a Peach™ oarlock force-sensing system. We hypothesize that the single IMU system, in conjunction with the UWT processing method, will provide kinematic measures of drive and stroke time consistent with the Peach^TM^ multi-sensor system. This work will support the use of single hull-mounted IMU sensors to reliably replace more cumbersome measurement tools for sub-stroke kinematic feature detection. 

## 2. Materials and Methods

### 2.1. Participants

Fourteen participants across three different boat classes were involved in this study; women’s eight coxed (W8+) (*n* = 9), women’s four non-coxed (W4−) (*n* = 4) and women’s single scull (W1x) (*n* = 1) groups belonging to a national rowing program were convenience sampled for this analysis, a sample in line with similar analyses for this sport and competition level [4] (mass = 78.77 ± 8.02 kg, height = 176.11 ± 7.64 cm, age = 28.27 ± 4.64). A power analysis was conducted using GPower ^TM^ software (version 3.1.9.3), where an approximate total sample size of 24 was calculated based on a power of 0.9, an alpha of 0.05 and a calculated effect size of 0.8 for paired comparisons across 3 groups (strokes from W8+, W4− and W1x). A total of 585 strokes were recorded (W8+ = 221 strokes, W4− = 183 strokes, W1x = 181 strokes). Ethical approval was obtained from the University of Victoria’s Human Research Ethics Board (ethics protocol number 24-0238). This study complied with the principles outlined in the Declaration of Helsinki. 

### 2.2. Testing Protocol

The subjects performed a single regularly scheduled race simulation training session (W8+ 2000 m session, W4−/W1x 1500 m session), performing at a wide variety of speeds and stroke rates (self-selected by the athletes for the fastest simulated race time) across the three boat classes, as outlined in Table 1. 

The distances for data collection were determined by the coaching staff and organized to simulate a race-like performance. For all sessions, the boats were equipped with the Peach^TM^ (UK) measurement system, which provided 50 Hz horizontal force data from each oarlock, as well as aperiodic stroke-by-stroke data including the drive time for each oarlock instrumented on the hull. A single AdMos^TM^ sensor was fastened to the boat hull near the stern (Figure 1) (Insiders, Lausanne, Switzerland) measuring tri-axial hull acceleration at 200 Hz (+/−16 g). The sensor was aligned with the x-axis along the translational axis of boat motion. This location was selected to provide a direct measure of boat hull acceleration, regardless of athletes’ position or orientation to the water, as well as to ensure that no interference with the athletes’ performance was possible. The drive and stroke times derived from the AdMos^TM^ IMU were compared to measures obtained by the Peach^TM^ system. 

### 2.3. Data Analysis 

The AdMos^TM^ IMU sensor data were exported using the Insiders^TM^ web-downloader native software. Force–time and drive time measurements were collected and exported using the PowerLine ^TM^ software (Version 4.12.00, Peach Innovations, Cambridge, UK). All of the data were analysed using custom-written Python™ (Python 3.11, Beaverton, OR, USA) software.

#### 2.3.1. IMU Stroke Feature Detection 

The forward acceleration axis of AdMos^TM^ IMU data were decomposed using a 9-level biorthogonal 4.4 (bior 4.4) undecimated wavelet transformation (UWT). The fourth approximation coefficient (cA3) was then negated, and peaks were detected to identify the minima of the acceleration corresponding to the start of the drive phase (D_S_). These indices were then used to segment the race session into individual strokes; the cA3 for the stroke was reversed, and the first peak was detected to identify the start of the cycle (C_S_), as defined by Klesnev 2010, which was then used as the onset of the stroke for all further analyses [6]. The third approximation coefficient (cA2) of the bior 4.4 wavelet was segmented using C_S_, each stroke was reversed, and the first peak detected was used to identify the drive end (D_E_), indicated by the minima observed after the peak acceleration in the drive phase (see Figure 2 for algorithm flowchart). It is important to note that this approach is based upon a 16-bit resolution accelerometer (±16 g) signal collected at 200 Hz, and any changes in these sensor parameters may require a unique UWT algorithm.

The drive time was then calculated as the difference in time between D_E_ and D_S_.
$$D_{T}=D_{E}-D_{S}$$

Finally, stroke time (ST) was defined as the difference between consecutive $D_{E}$ values, indicating the elapsing time between strokes. 

#### 2.3.2. Force Trace Feature Detection 

Force–time data from the Peach^TM^ system were used to determine stroke time. All horizontal forces (F_GX_) from each oarlock gate were summed, estimating the total horizontal force of the system (i.e., the rowing hull, athletes and equipment) as a whole (F_Tx_).
$$F_{Tx}=\sum_{n}F_{1x}+F_{2x}+\ldots \;F_{nx}$$

The total force was then used to evaluate the stroke time, defined as the consecutive difference between force-derived drive-end (D_EF_) features. To determine the drive end from the force trace, force peaks were first identified for each stroke, using simple peak detection. From here, the drive end was then defined as the first zero crossing (positive to negative) following the force peak. Consecutive time stamps from each drive end were then subtracted to give the force-derived stroke time ST_F_.
$${ST}_{F(n)}=D_{EF(n)}-D_{EF(n-1)}$$

Stroke time measures were then compared to those obtained using a similar detection technique with the AdMos^TM^ sensor. See Figure 3. 

#### 2.3.3. Peach™ Drive Time

The drive time from the Peach™ system was obtained using its proprietary algorithm. This algorithm uses threshold crossings on the force trace to define the drive start and drive end for individual oarlocks present in the system, producing a drive time measure for each oarlock gate equipped on the rowing hull. To determine the system drive time for each boat class, drive time values were averaged across all oarlocks for each stroke. This average drive time was used as the criterion measure for analysis to compare against the drive time obtained using the AdMos^TM^ sensor.

### 2.4. Statistical Analysis

To evaluate the reliability of the drive times and stroke times calculated using the UWT accelerometer data from AdMos^TM^ IMU sensors compared to the ecological standard Peach™ measures, intra-class correlation analyses for one-way random effects using average random raters (ICC [2K]) [12] were completed. Significance was set at α < 0.05 for all comparisons, and correlation coefficients were interpreted as <0.50 = poor, 0.50–0.75 = moderate, 0.75–0.90 = good and >0.90 = excellent [12]. Further, a Bland–Altman analysis was used to evaluate the bias and agreeability between signals. Finally, a regression analysis was used as a further means of representing the agreement between signals. 

## 3. Results

### 3.1. Intra-Class Correlation Analysis

The ICC [2K] results are shown in Table 2. For all measures, the AdMos^TM^ sensor-derived metrics displayed very strong or strong reliability when compared to the Peach^TM^ system reference. 

### 3.2. Bland–Altman Analysis

A Bland–Altman analysis identified a very minor bias between the values derived from the AdMos^TM^ and Peach^TM^ systems. For the drive time, W8+, W4− and W1x displayed biases of −0.005, −0.014 and −0.015 s, respectively, with an average bias of −0.011 s across boat classes. The stroke time measures produced similar biases, with values of −0.0005, −0.0115 and −0.0005 for the W8+, W4− and W1x classes, respectively, generating an average bias of −0.0042 s across boat classes. See Figure 4.

### 3.3. Regression Analysis

A linear regression analysis was applied to the measures to observe trends in the relationships between measures derived from each measurement system. The slopes, intercepts and r^2^ values from each regression were determined for each measure across boat classes. All regression models returned significant values (*p* < 0.05), indicating linear relationships between measures. Correlation values (r^2^) of 0.819, 0.782 and 0.731 were determined for the drive time measures for the W8+, W4− and W1x classes, respectively. For stroke time, similar correlations were observed, with r^2^ values of 0.973, 0.883 and 0.830 for W8+, W4− and W1x. See Figure 5.

## 4. Discussion

This is the first study to implement a UWT analysis to detect rowing kinematic phases based on a single rowing hull-mounted accelerometer sensor signal across boat classes. The present approach has excellent agreement against the standard force-derived algorithm and provides a reliable alternative that requires no modifications between boat classes. While the determination of drive start can be implemented using simple peak detection algorithms, this approach is able to consistently and accurately determine the end of the drive, which, for the first time, allows for important segmentation of the drive and recovery phases in rowing using a single hull-mounted accelerometer. 

The standard measurement approach for determining and segmenting kinematic and kinetic features in rowing is through the instrumentation of the Peach™ system. This system uses a series of sensors providing many measures, including oarlock force, oar angle, foot-stretcher force and hull kinematics, obtained from GPS and accelerometry devices. An important consideration in multi-sensor systems is the use of measures from different sensors to categorize the unique aspects of rowing kinematics and kinetics. For example, drive duration is often determined through either the transitions in the oar angle or the rise and fall of the oarlock horizontal force. While the oar angle is commonly used [3,6,13], the Peach™ system uses oarlock force thresholding as a more reliably determined signal characteristic. For this reason, in the present study, we compared the drive time calculated from the UWT of the hull accelerometer signal to the proprietary drive times calculated by the Peach™ system. This enabled a direct comparison to an established standard. In comparison to the Peach™ proprietary measures, we found a good (W1x) to excellent (W4−, W8+) ICC with our novel UWT approach, which supports the use of our algorithm for the determination of both the drive start and the drive end. Further, this study also compared the stroke time using the oarlock force signal for the determination of the drive end as the first negative zero crossing following peak force. This feature in the data was chosen as the force zero crossing aligns temporally with the drive end of accelerometer data and would allow for a direct temporal comparison. This comparison demonstrates the excellent to near-perfect correlation between the UWT acceleration drive-end determination and the force drive-end calculation. Taken together, these results demonstrate that while this algorithm can validly and reliably determine the drive end, as compared to the force signal analysis, differences in the determination of the drive start related to the proprietary Peach™ approach and/or based on the nature of the force signal may require further refinement of the drive start feature detection. This may rely on a greater understanding of the rowing stroke cycle and, specifically, how the drive timing and other important rowing phases are determined. 

Understanding the biomechanics of the rowing stroke and its relationship to temporal phases is well presented by Kleshnev (2010), who identified nine microphases of the rowing stroke that can be used to support biomechanical feedback [6]. These microphases were identified based on the centre-of-mass accelerations of the boat, the rower and the global system (i.e., rower, boat, oars, rigging, etc.). These kinematics were considered related to the kinetic energy of the rower–boat system and reflect the effectiveness of the rowing technique [6]. The microphases identified by Kleshnev (2010) strongly support an acceleration-based approach to the determination of rowing kinematics [6]. This positions the present UWT-based technique as a foundational approach to quantifying rowing phase transitions and the resulting acceleration profiles within these phases. For example, while a single hull-mounted accelerometer may not be sufficient alone, the simultaneous use of a hull- and athlete-mounted sensor, alongside the use of the UWT, may best support the analysis proposed by Kleshnev (2010) [6]. Another important consideration is that there is presently a lack of consistency with respect to what feature and what signal should be used to determine each phase. The universal adoption of an acceleration-based approach to rowing technique segmentation, as presented by Kleshnev (2010), may help to support greater consistency in both research and coaching and allow for a more aligned comparison in rowing research [6]. For example, Holt et al. (2021) examined distinctive peaks and valleys in a hull-mounted accelerometer signal and compared these signal features to boat velocity [2]. Their findings showed that specific changes in acceleration and jerk features could be related to changes in overall performance and are potentially useful as feedback metrics. While Holt et al. (2021) used manually tuned lowpass filters and peak/valley detection to identify these landmarks and determine the drive onset and offset values within the acceleration profile, our method utilizes the wavelet coefficients that contain the frequency bands of these features, providing a signal that is more consistent across variable speeds and boat classes [2]. 

Wavelet analyses, such as the UWT technique outlined in this study, are commonly utilized in biological signal processing for the de-noising and feature detection of signals that share similar characteristics to the rowing acceleration profile. For example, electrocardiogram (ECG) PQRST complex feature detection often employs the UWT to locate signal peaks and valleys associated with phases of the cardiac cycle because the translation invariance of the decomposition allows for accurate temporal feature locations with time-varying signals such as heart rate, similar to what is observed with the variable nature of stroke rate across rowing boat classes and within the race session [14,15]. For example, Peressutti et al. (2010) used discrete wavelet transformations to analyse heart rate and heart rate variability during training in place of standard filtering and detrending techniques [9]. Undecimated wavelet transformation is particularly useful for signal feature detection, as the transformation is shift-invariant, avoiding downsampling of the signal, and utilizes zero-phase filters, which maintain the temporal features of the transformed signal [10]. For rowing, the stroke rate can vary significantly based on the boat class and specific race strategy or training intention. This requires a method of stroke feature identification that can reliably detect features of a signal with varying feature frequency, which we have demonstrated with our UWT method. 

Inertial sensors represent a cost-effective and simple means of measuring boat acceleration and calculating stroke characteristics, compared to the more invasive processes involved with force instrumentation. However, the use of a single sensor can be limiting, and the determination of sub-features relies heavily on robust and reliable processing methods to ensure the accuracy of these indirect measures. The UWT method outlined in this study achieves the goal of providing a reliable and accurate means of determining drive and recovery stroke metrics using a single hull-mounted accelerometer; however, further investigation is required to determine the optimal instrumentation for measuring more specific stroke mechanics such as blade extraction and oar gate synchronization. In addition, the use of an IMU with a different sample rate or measurement range could require the selection of different approximation coefficients than those detailed in our proposed UWT algorithm. Finally, the current study is limited in that only female athletes across three boat classes were recruited for this study. Further research exploring other boat classes across genders may be required. 

## 5. Conclusions

In summary, this study strongly supports the use of a single hull-mounted IMU sensor to determine the temporal and kinematic sub-stroke performance measures of drive and stroke time. This single sensor and accompanying UWT processing technique can provide accurate and reliable measures across multiple boat classes without the need for force data, making the acquisition of stroke cycle data linked to performance indicators more accessible and readily available for coaches, practitioners and athletes.

## Figures and Tables

**Figure 1 sensors-24-06085-f001:**
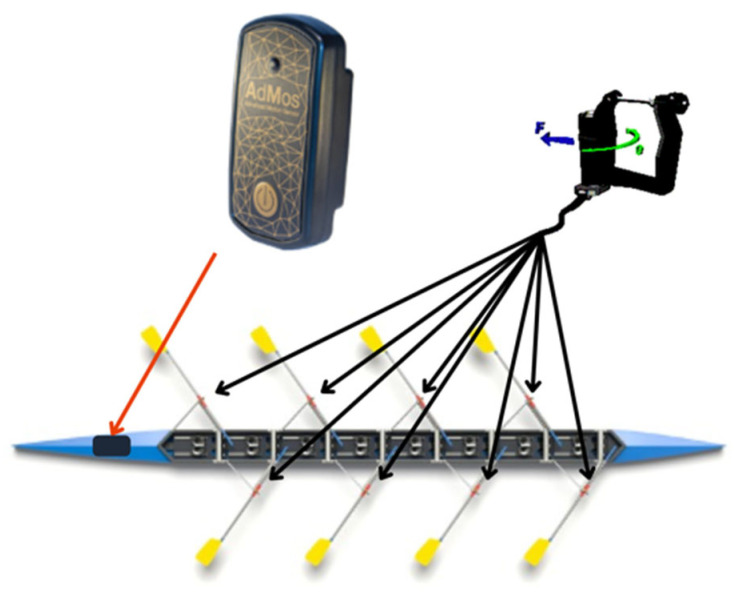
The mounting location of the AdMos^TM^ sensor on the boat hull. The location and orientation of the sensor are identified by the black oval present near the stern of the visualized boat above (location shown using the red arrow). Also shown is the location of the force-sensing oarlocks, which are part of the Peach^TM^ measurement system (location shown using the back arrows).

**Figure 2 sensors-24-06085-f002:**
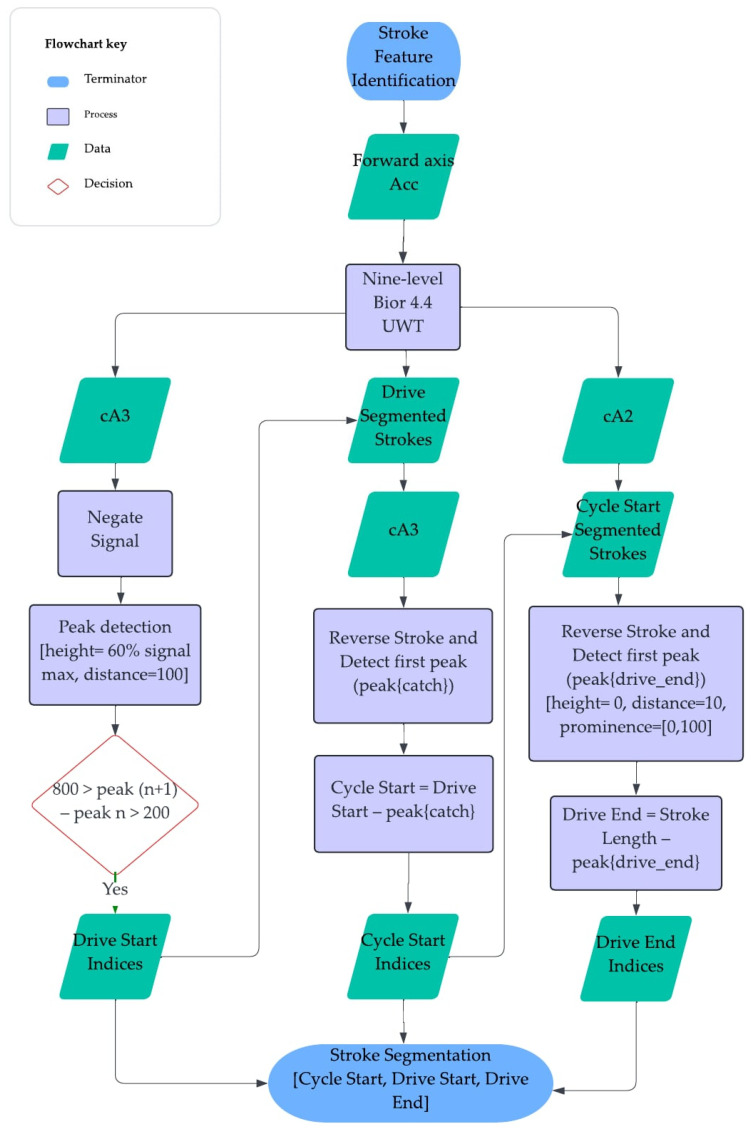
The stroke feature detection algorithm flow chart detailing the detection of the drive and recovery phases of the rowing stroke.

**Figure 3 sensors-24-06085-f003:**
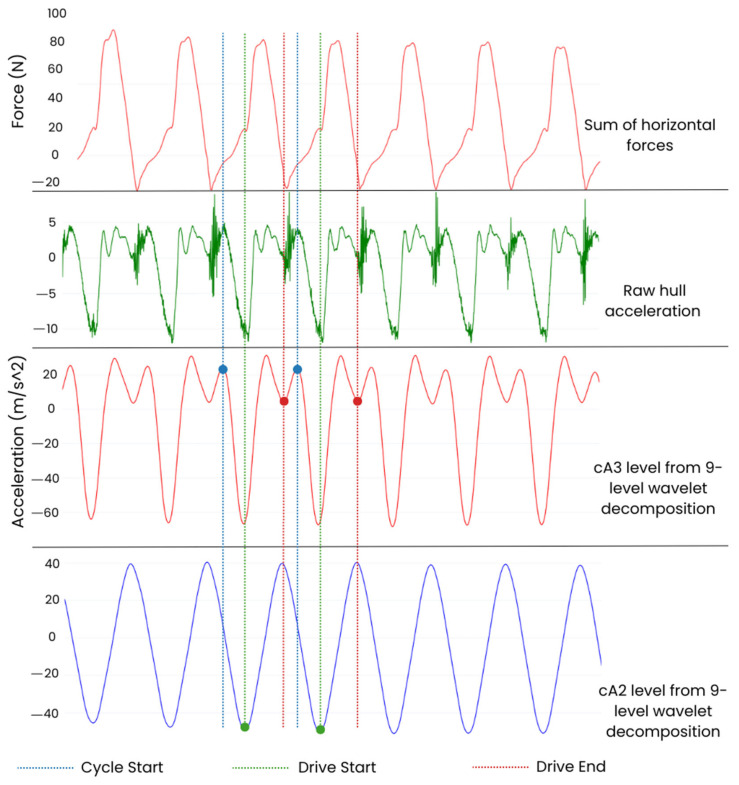
A visualization of the various decomposition levels used to detect features using the acceleration signal. The horizontal oarlock force (top, red) from consecutive strokes is time-aligned with the corresponding forward rowing acceleration signal (second from top, green). The third (red) and fourth (blue) plots indicate the fourth (cA3) and third (cA2) UWT levels of a nine-level biorthogonal 4.4 decomposition applied to the forward boat acceleration. The features used to define drive start, drive end and cycle start are identified across signals (force and acceleration). Drive start is shown with the vertical green dotted line, drive end is shown with the vertical red dotted line and cycle start is shown with the vertical blue dotted line. The specific features detected with the algorithm are identified by larger markers and colour-coded for each feature.

**Figure 4 sensors-24-06085-f004:**
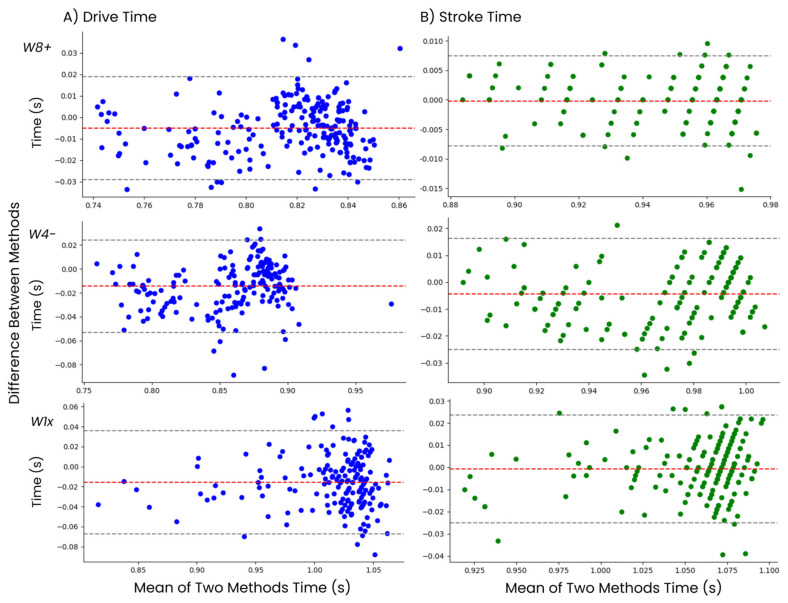
The Bland–Altman analysis comparing AdMos^TM^ sensor-derived measures to metrics from the Peach^TM^ system. (**A**) compares measures of drive time, and (**B**) compares measures of stroke time. All measures are separated by respective boat class, indicated as rows.

**Figure 5 sensors-24-06085-f005:**
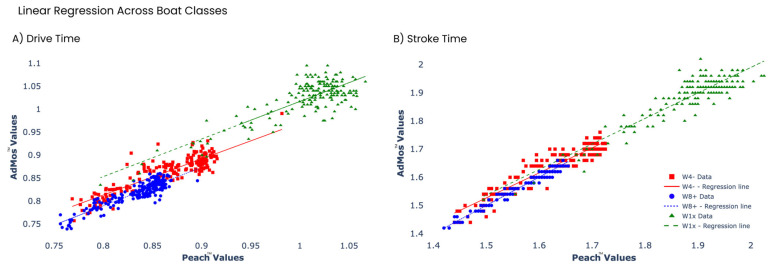
The linear regression analysis comparing the Peach™ and AdMos^TM^ sensors for drive (**A**) and stroke times (**B**). Different boat classes are separated by marker shape (square: W4−, circle: W8+ and triangle: W1x) and colour, as indicated in the legend.

**Table 1 sensors-24-06085-t001:** Averages and standard deviation values for stroke rate and average stroke velocity of analysed dataset separated by boat class.

Boat Class	Stroke Rate (SPM)	Average Stroke Velocity (m/s)
W8+	37.88 ± 1.46	5.47 ± 0.19
W4−	36.72 ± 1.90	4.92 ± 0.26
W1x	31.90 ± 1.61	4.09 ± 0.21

**Table 2 sensors-24-06085-t002:** Intra-class correlation values using average random raters (ICC (2k)). Drive and stroke time measures were compared between the Peach^TM^ and AdMos^TM^ sensors. Values highlighted with an asterisk (*) or double asterisk (**) represent good or excellent agreeability, respectively.

Measure	ICC	F	Df	CI 95%
W8+				
Drive Time	0.937 **	22.890	221	0.89, 0.96
Stroke Time	0.993 **	146.572	221	0.99, 0.99
W4−				
Drive Time	0.901 **	15.472	183	0.70, 0.95
Stroke Time	0.963 **	31.397	183	0.94, 0.98
W1x				
Drive Time	0.881 *	11.055	181	0.73, 0.94
Stroke Time	0.954 **	21.505	181	0.94, 0.97

## Data Availability

The data used in this study were shared by Rowing Canada Aviron and are not available for further sharing.

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
