# Peer review of "The Determination of On-Water Rowing Stroke Kinematics Using an Undecimated Wavelet Transform of a Rowing Hull-Mounted Accelerometer Signal"

_sensors, 2024, doi:10.3390/s24186085_

Round 1
Reviewer 1 Report
Comments and Suggestions for Authors
Dear authors, it has been a pleasure to review your paper entitled: "Determination of On-water Rowing Stroke Kinematics Using an Undecimated Wavelet Transform of a Rowing Hull mounted Accelerometer Signal".
The paper is well written and presents significant findings applicable to the field of study. However, there are a number of considerations regarding the methodology that need to be corrected and resolved.
Participants:
- Why was the study made up of only women? Is there any aspect or study that shows gender differences in rowing that could be relevant to comment on?
-It is necessary to provide how the sample size was calculated.
-It would be interesting to include a descriptive table with the characteristics of the sample (age, weight, height, sports level, etc.)
- This section should indicate how the sample was recruited, indicating the inclusion/exclusion criteria. In addition, this section should include the corresponding ethics committee number, whether the subjects signed the informed consent and whether the Helsinki Declaration was followed.
Testing Protocol:
- On what basis were the distances selected in the protocol decided?
-It is mentioned that the protocol was developed at different speeds and stroke rates. These values ​​should be specified and the reason for their choice. A table with these values ​​would be interesting.
-The AdMos IMU must be described in greater detail, specifying its characteristics. The reason for choosing its location on the boat must also be justified.
Discussion:
- Are there any limitations of the study that deserve comment?
Reviewer 2 Report
Comments and Suggestions for Authors
Determination of On-water Rowing Stroke Kinematics Using an Undecimated Wavelet Transform of a Rowing Hull mounted Accelerometer Signal REVIEW
This study aimed to is investigate usefulness and efficiency of usage of undecimated wavelet transform technique in detecting individual features in the stroke acceleration profile from a single rowing hull-mounted accelerometer.
The use of advanced technology as well as development of innovative way in its usage in analysing sport performance the length of the study is main strength of this study, especially when level of rowing sport world vide is taken into account.
Subject sample which is small and not representative are main weakness of this paper. Although significant number of strokes were analysed, generalizability of findings is questionable, since we have only 18 participants and all women.
I advise authors to give in more details how and why they made study sample like this (power analyses, inclusion-exclusion criteria, sampling method...).
Reference number 12 is not the best choice to support author’s choice of interpretations, I suggest to find some more appropriate, there are a lot of them.
Study design, data gathering and analysing are appropriate.
The explanation of the coordinate system in the figure 4 is missing.
Thanks to methodologically rigorous data collection process, this study gives in detail description of relationships among two ways of detecting individual features in the stroke acceleration profile. In this way it gives us clear answer to a research question, but also opens the space for technological improvement in practise.
Because the main disadvantage of the research is a small sample I advise the authors to compare it with similar studies. Authors in their next studies should have bigger more divers and representative sample as well as comparison of different ways of programming the training load as well as their systematic experimental variation.
Results were correctly analyzed and valid conclusions were drawn from them. These studies provide not only a scientific but also a significant practical contribution.
Except reference 12 the references are appropriate.
I do not see informed consent of participants to take part in the study.
I have not additional comments.
